# Structure-based design and characterization of Parkin-activating mutations

Michael U Stevens[1],*, Nathalie Croteau[2,3],*, Mohamed A Eldeeb[4],*, Odetta Antico[1], Zhi Wei Zeng[2], Rachel Toth[1], Thomas M Durcan[4], Wolfdieter Springer[5,6], Edward A Fon[4], Miratul MK Muqit[1], Jean-François Trempe[2,3]

**Autosomal recessive mutations in the Parkin gene cause Parkinson's disease. Parkin encodes an ubiquitin E3 ligase that functions together with the kinase PINK1 in a mitochondrial quality control pathway. Parkin exists in an inactive conformation mediated by autoinhibitory domain interfaces. Thus, Parkin has become a target for the development of therapeutics that activate its ligase activity. Yet, the extent to which different regions of Parkin can be targeted for activation remained unknown. Here, we have used a rational structure-based approach to design new activating mutations in both human and rat Parkin across interdomain interfaces. Out of 31 mutations tested, we identified 11 activating mutations that all cluster near the RING0:RING2 or REP:RING1 interfaces. The activity of these mutants correlates with reduced thermal stability. Furthermore, three mutations V393D, A401D, and W403A rescue a Parkin S65A mutant, defective in mitophagy, in cell-based studies. Overall our data extend previous analysis of Parkin activation mutants and suggests that small molecules that would mimic RING0:RING2 or REP:RING1 destabilisation offer therapeutic potential for Parkinson's disease patients harbouring select Parkin mutations.**

## Introduction

Mutations in the *PRKN* (Kitada et al, 1998) and *PINK1* (Valente et al, 2004) genes account for most of the recessive early-onset Parkinson's disease (PD) cases (Martin et al, 2011). The *PRKN* gene encodes Parkin, an E3 ubiquitin (Ub) ligase that mediates a mitochondrial quality control pathway that is also dependent on the PINK1 kinase activity (Bayne & Trempe, 2019). Evidence in support of this hypothesis initially came from genetic models in *Drosophila*, where ablation of either gene causes muscle degeneration and male sterility as a result of mitochondrial defects (Greene et al,

2003; Clark et al, 2006; Park et al, 2006). Although *PRKN* KO mice do not display neuron loss (Goldberg et al, 2003), they exhibit deficits in striatal neuron excitability and perturbation in mitochondrial proteins and respiration (Palacino et al, 2004; Periquet et al, 2005). Genetic crosses of *PRKN* KO with the *mutator* mouse, which has a defective mitochondrial DNA polymerase, show nigrostriatal degeneration and inflammation, consistent with Parkin having a role in protecting dopaminergic (DA) neurons from mitochondrial damage (Pickrell et al, 2015; Sliter et al, 2018). Likewise, *PINK1* KO mice also do not display neurodegeneration, but similar to *PRKN* KO exhibit marked mitochondrial defects (Gautier et al, 2008). Furthermore, *PINK1* KO mice subjected to bacterial infections undergo loss of DA axonal varicosities, as a result of mitochondrial antigen presentation and activation of cytotoxic T cells in the brain (Matheoud et al, 2019). Thus, loss of Parkin or PINK1 proteins results in mitochondrial dysfunction, which can ultimately lead to neurodegeneration. It has also been reported that the levels of soluble Parkin decrease with age and PD patients show elevated levels of insoluble Parkin, which would be associated with reduced biochemical activity (Pawlyk et al, 2003; Wang et al, 2005; Tokarew et al, 2021). Therefore, enhancing Parkin's intrinsic activity may have beneficial therapeutic implications.

The mechanism of action of Parkin and PINK1 has been elucidated in the last decade. In cultured cells, the two proteins mediate selective mitochondrial autophagy (mitophagy) after chemical-induced mitochondrial depolarization, accumulation of misfolded mitochondrial proteins or after exposure to complex I inhibitors (Narendra et al, 2008, 2010; Geisler et al, 2010; Matsuda et al, 2010; Jin & Youle, 2013). PINK1 is a kinase stabilised on damaged mitochondria, where it phosphorylates ubiquitin at Serine65 (Kazlauskaite et al, 2014b; Kane et al, 2014; Koyano et al, 2014). The resulting phospho^Ser65-ubiquitin (pUb) acts as a receptor for Parkin to be recruited to sites of damaged mitochondria, thus enabling PINK1 to phosphorylate Parkin, which in turn dramatically increases its E3 ubiquitin ligase activity (Kondapalli et al, 2012). Active Parkin ubiquitinate substrates on the outer mitochondrial membrane

[1]MRC Protein Phosphorylation and Ubiquitylation Unit, School of Life Sciences, University of Dundee, Dundee, UK  [2]Department of Pharmacology & Therapeutics, McGill University, Montréal, Canada  [3]Centre de Recherche en Biologie Structurale, Montpellier, France  [4]McGill Parkinson Program, Montreal Neurological Institute, McGill University, Montreal, Canada  [5]Department of Neuroscience, Mayo Clinic, Jacksonville, FL, USA  [6]Neuroscience PhD Program, Mayo Clinic Graduate School of Biomedical Sciences, Jacksonville, FL, USA

Correspondence: ted.fon@mcgill.ca; m.muqit@dundee.ac.uk; jeanfrancois.trempe@mcgill.ca
*Michael U Stevens, Nathalie Croteau, and Mohamed A Eldeeb contributed equally to this work

such as Mitofusin, Miro, VDAC, etc., which are modified at different rates and extents (Tanaka et al, 2010; Chan et al, 2011; Ordureau et al, 2018; Antico et al, 2021; Vranas et al, 2022). The accumulation of the ubiquitinated proteins is rapid because of a feedforward mechanism and has mainly been linked to mitophagy (Ordureau et al, 2014). Beyond mitophagy, Parkin/PINK1 can also mediate mitochondrial arrest in neurons (Wang et al, 2011), spatially restricted mitophagy (Yang & Yang, 2013), and mitochondria-derived vesicle formation (McLelland et al, 2014). Parkin/PINK1 most likely mediate stress-evoked degradation of a subset of mitochondrial proteins, as evidenced by turnover measurements in *Drosophila* (Vincow et al, 2013), and which is distinct from the regulation of basal mitophagy that is independent of PINK1 in mice (McWilliams et al, 2018b). Furthermore, pUb, which can be detected in human and rodent brains (Fiesel et al, 2015), is absent in PINK1-null mammals and reduced in Parkin-null or Parkin[S65A] mutant knock-in mice, and is increased in the mutator mouse and sporadic PD patients (Pickrell et al, 2015; McWilliams et al, 2018a; Hou et al, 2018; Watzlawik et al, 2021). These observations strongly support PINK1 being the primary activator of Parkin in vivo, with pUb serving as a marker for mitochondrial damage.

Structural studies have unveiled fundamental details regarding Parkin activation at the molecular level. Parkin belongs to the RING-in-Between-RING family of E3 Ub ligases, which transfers ubiquitin to a substrate from an E2 ubiquitin-conjugating enzyme via formation of a thioester intermediate on a reactive cysteine (Parkin Cys431) in the RING2 catalytic domain (Wenzel et al, 2011). Structural and biochemical analysis of Parkin showed that the protein is auto-inhibited in the basal, unstimulated state (Chaugule et al, 2011; Riley et al, 2013; Spratt et al, 2013; Trempe et al, 2013; Wauer & Komander, 2013). The ubiquitin-like (Ubl) domain and repressor element of Parkin (REP) both occlude the E2-binding site on RING1, and Cys431 is sequestered at the interface of RING0 and too far from the E2~Ub conjugate for efficient transfer. Upon binding to pUb, the Ubl domain is released from RING1 and can be freely phosphorylated by PINK1 at Ser65 (Kazlauskaite et al, 2015; Kumar et al, 2015; Sauvé et al, 2015; Wauer et al, 2015). The phospho-Ubl domain then binds to RING0 and competes with the RING2 domain, which dissociates concomitantly with the REP, thus allowing E2~Ub binding (Condos et al, 2018; Gladkova et al, 2018; Sauvé et al, 2018). Although there is no crystal structure of the resulting Parkin thioester transfer complex, how the RING2 domain will interact with E2~Ub, can be inferred from the structure of the RING-in-Between-RING ligase HOIP bound to UbcH5B ~ Ub (Lechtenberg et al, 2016). Once Ub is charged on Cys431, it is transferred to a substrate lysine in a reaction catalysed by His433 (Vranas et al, 2022). The broad range of Parkin substrates suggests that is unlikely to recognize a specific substrate motif and substrate specificity is rather conferred by proximity at sites of activation.

In the light of these in vitro studies, it appears that interdomain interactions maintain Parkin in an auto-inhibited state. Indeed, mutagenesis of residues located at inhibitory interfaces, such as Phe146 in RING0 or Trp403 in the REP, result in a dramatic increase in the ligase activity of Parkin and mitochondrial recruitment (Trempe et al, 2013), and can rescue S65A or ΔUbl Parkin in cis, in both ubiquitination and mitophagy assays (Sauvé et al, 2015; Tang et al, 2017). Synthetic F146A and W403A mutations also rescue seven

PD-associated Parkin missense mutations that disrupt Parkin activity through various mechanisms (Yi et al, 2019). On the other hand, the N273K mutation in RING1, which repels the Ubl domain, accelerates mitochondrial recruitment but does not rescue the S65A mutation (Tang et al, 2017). Because activating mutations can result in the constitutive activation of Parkin in the cytosol, it is conceivable that strongly activating mutations may induce rapid Parkin turnover via auto-ubiquitination, which would reduce the steady-state levels of the protein in cells and thus reduce its actual activity towards mitochondrial substrates. Another factor to consider is the effect of the mutation on the solubility of Parkin; for instance, we have found that mutations at the RING0:RING2 interface reduce the solubility of the protein in vitro (Tang et al, 2017). Finally, only a small subset of interfaces and mutants has so far been tested, each of which may tune Parkin's activity and stability to various degrees. Thus, a systematic and comprehensive assessment of activating mutations is required.

Here, we have designed and introduced 31 synthetic mutations in mammalian Parkin as candidates for activation, and have characterized their effects on recombinant protein expression, thermal stability, and ubiquitination activity in vitro. We identified 11 mutations that activate both rat and human Parkin. We find that all activating mutations cluster near the RING0:RING2 or REP:RING1 interfaces and reduce the thermal stability of Parkin. These mutations were introduced in human cells and tested for their ability to enhance depolarization-induced mitophagy using a fluorescent-reporter assay. From this group of 11, we identified two novel mutations (V393D and A401D) that robustly enhance mitophagy and rescue the defective S65A mutation, in addition to the W403A mutation previously reported. These mutations provide a molecular basis for future studies in neuronal cells and a map to a "hot-spot" of activation in Parkin that will serve as a scaffold for the design of small-molecule activators.

## Results

### Survey and design of activating mutations in Parkin

Different groups have reported mutations that activate Parkin in vitro (Table 1). These include mutations at the RING0:RING2 and REP:RING1 interfaces (F146A, W403A, F463Y, F463A) (Riley et al, 2013; Trempe et al, 2013; Wauer & Komander, 2013), and residues within the RING0:RING1 interface that remodels upon phosphorylation (H227N, E300A/Q) (Kumar et al, 2015). Mutations in RING1 residues that bind to the Ubl (N273K, L266K/R) also activate Parkin (Kumar et al, 2015; Sauvé et al, 2015; Wauer et al, 2015). Although mutations in the Ubl at the corresponding interface also activate Parkin in vitro (I44A, H68A, etc.) (Kumar et al, 2015), these mutants are likely to interfere with PINK1 phosphorylation (Wauer et al, 2015; Rasool et al, 2018), and were avoided. The PD mutation R33Q has also been shown to activate Parkin (Chaugule et al, 2011).

In addition to the mutations reported above, we also sought to design new mutations that may increase Parkin's activity based on structures of inactive and active mammalian Parkin (Fig 1). Because human His227 is an Asn in rat and mouse Parkin, this residue was mutated to Ala as well (Table 1). This residue is located in proximity

**Table 1. Proposed activating mutations in Parkin (conserved in rat and human Parkin).**

| Mutation | Domain | Reference(s) |
|---|---|---|
| R33Q | UBL | Chaugule et al (2011) |
| Y143D | RING0 | This article |
| Y143E | RING0 | This article |
| F146A | RING0 | Trempe et al (2013); Ordureau et al (2014); Tang et al (2017); Yi et al (2019) |
| W183Y | RING0 | This article |
| F208Y | RING0 | This article |
| H227N (human only) | RING1 | Kumar et al (2015) |
| H/N227A (human/rat) | RING1 | This article |
| L266K | RING1 | Sauvé et al (2015) |
| L266R | RING1 | This article |
| N273K | RING1 | Sauvé et al (2015); Tang et al (2017) |
| E300A | RING1 | Kumar et al (2015) |
| E300Q | RING1 | Kumar et al (2015) |
| E370A | IBR | This article |
| V393D | REP | This article |
| V393K | REP | This article |
| A397D | REP | This article |
| A397K | REP | This article |
| A398T | REP | Periquet et al (2003) |
| A398D | REP | This article |
| A398K | REP | This article |
| A401D | REP | This article |
| A401K | REP | This article |
| W403A | REP | Trempe et al (2013); Ordureau et al (2014); Sauvé et al (2015); Tang et al (2017); Yi et al (2019) |
| W403F | REP | This article |
| W462A | RING2 | This article |
| W462F | RING2 | This article |
| W462Y | RING2 | This article |
| F463Y | RING2 | Riley et al (2013) |
| D464A | RING2 | This article |
| D464K | RING2 | This article |

to Trp183 at the RING0:RING1 interface, and thus, we also introduced the W183Y mutation, a conservative mutation given that Trp183 is also part of the core of the RING0 domain. Given that the REP W403A mutant displays some of the most desirable characteristics (stable, soluble, enhances Parkin recruitment, and mitophagy), we introduced multiple additional REP mutations (Fig 1). In particular, we mutated small aliphatic residues in the REP helix to charged residues to disrupt their interactions with the RING1 domain; namely residues Val393, Ala397, Ala398, and Ala401 (Fig 1).

Intriguingly, A398T was discovered in a single PD patient as a heterozygous mutation (Periquet et al, 2003), but it is unclear whether it is pathogenic. It was also reported to activate Parkin in vitro (Wauer & Komander, 2013), and we therefore included this mutation in our panel. In addition, we mutated Trp403 to a Phe to potentially achieve a lower degree of activation compared with W403A. We noted that Trp462 in RING2 is also mediating an interaction with Phe208 in RING0, and therefore mutated these aromatic residues to Tyr to perturb the interaction but maintain aromaticity as these residues are partially buried in the core of the domains (Fig 1). Finally, two groups reported that Abl phosphorylates Parkin on Tyr143, which was reported to inactivate the enzyme (Ko et al, 2010; Imam et al, 2011). However, inspection of the rat apo Parkin structure suggests that introducing a negative charge at this position may induce an electrostatic repulsion with the C-terminus of RING2 and thus may result in the activation of Parkin (Fig 1). We thus introduced the Y143D and Y143E mutations.

All mutations mentioned above were introduced in both human and rat Parkin, to identify mutations that could later be studied in both rodent models and human cells. Furthermore, the experiments with rat and human Parkin were performed in two different labs using different plasmids for bacterial expression, to ensure robust identification of mutations that activate Parkin under independent purification and assay conditions. All mutants purified successfully with yields comparable to those obtained for the WT protein (Table S1).

**Activating mutants cluster at the REP:RING1 and RING0:RING2 interfaces**

We quantified the E3 ubiquitin ligase activity of human and rat Parkin mutants using a simple autoubiquitination assay of recombinant Parkin incubated with the E2 ubiquitin-conjugating enzyme UbcH7, and the E1 enzyme and ATP. While activating mutants such as W403A robustly autoubiquitinate, it was difficult to quantify Parkin by densitometry because free polyubiquitin chains also formed in the assay and migrated at the same molecular weight as unmodified Parkin (Fig 2A [human] and Fig S1 [rat]). However, we observed that activated Parkin also ubiquitinates UbcH7, whose SDS–PAGE migration is distinct from polyubiquitin chains. We therefore quantified Parkin activity by quantifying the loss of unmodified UbcH7 on SDS–PAGE, as reported by our group recently (Fava et al, 2019). All experiments were performed in duplicate and included WT and W403A variants on every gel as internal benchmark controls for low and high activities, respectively.

There was a strong agreement in the activation profile of mutants across human and rat assays (Fig 2B). The data confirm that F146A, W403A, and F463Y activate Parkin in both rat and human in vitro, as observed previously (Riley et al, 2013; Trempe et al, 2013). We also find that mutants Y143D, Y143E, V393D, A401D, and A401K activate Parkin to levels comparable to W403A in both species. The mutants V393K, A398D, and W403F activated Parkin to a lower extent in both species. Finally, we found that the A398T mutation in human Parkin had a minor effect, but a stronger effect in rat Parkin. All other mutations had minor or no effects on Parkin ligase activity. Overall, these experiments identify 11 mutations that consistently activate both rat and human Parkin. These mutations are all located

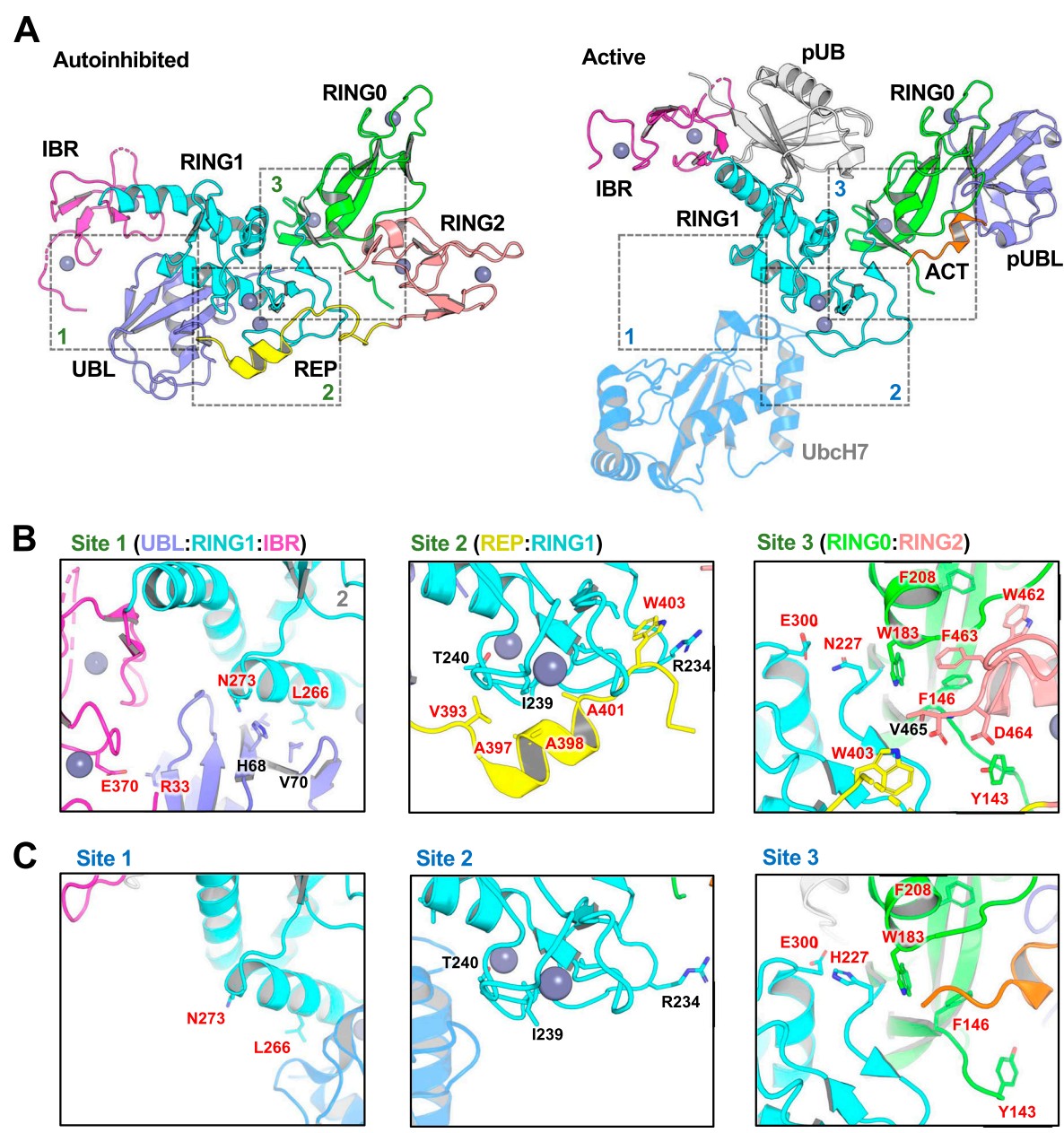

**Figure 1. Design of activating mutations in Parkin.**
**(A)** Cartoon representations of auto-inhibited apo rat Parkin (left; PDB ID: 4ZYN) and active human phospho-Parkin bound to phospho-Ub (right; PDB ID: 6GLC). The structure of auto-inhibited apo human Parkin (PDB ID 5C1Z, not shown) is nearly identical to the rat ortholog but lacks Y143. The position of UbcH7 is based on the superposition of activated fly phospho-Parkin bound to pUb and UbcH7 (PDB ID: 6DJW, not shown). The different domains of Parkin are shaded and labelled, and dashed boxes indicate interfaces between these domains. Note that the structures of activated Parkin lack the REP and RING2 domains. **(B)** Residues within the binding interfaces of auto-inhibited rat Parkin are highlighted, with residues mutated for this work labeled in red. **(B, C)** Residues within the same binding interfaces as in (B), but for the activated human phospho-Parkin:pUb complex.

at the RING0:RING2 or REP:RING1 interfaces (Fig 2C), suggesting these are most critical for auto-inhibition.

## Thermal stability assays reveal that destabilizing mutations activate Parkin, but only at the RING0:RING2 or REP:RING1 interfaces

Previous studies have suggested that Parkin could be activated through destabilization of its auto-inhibited state (Chaugule et al,

2011; Riley et al, 2013; Spratt et al, 2013; Trempe et al, 2013; Wauer & Komander, 2013; Caulfield et al, 2014). To investigate this hypothesis, we determined the stability of recombinant Parkin mutants using a fluorescence-based thermal shift assay (Fig 3A). WT rat and human Parkin have average melting temperatures ($T_m$) of 55.8°C and 59.0°C, respectively. Overall, all 11 activating mutations lower the $T_m$ of Parkin by 3–7°C compared with WT. To better understand the relationship between thermal stability and ubiquitin ligase activity,

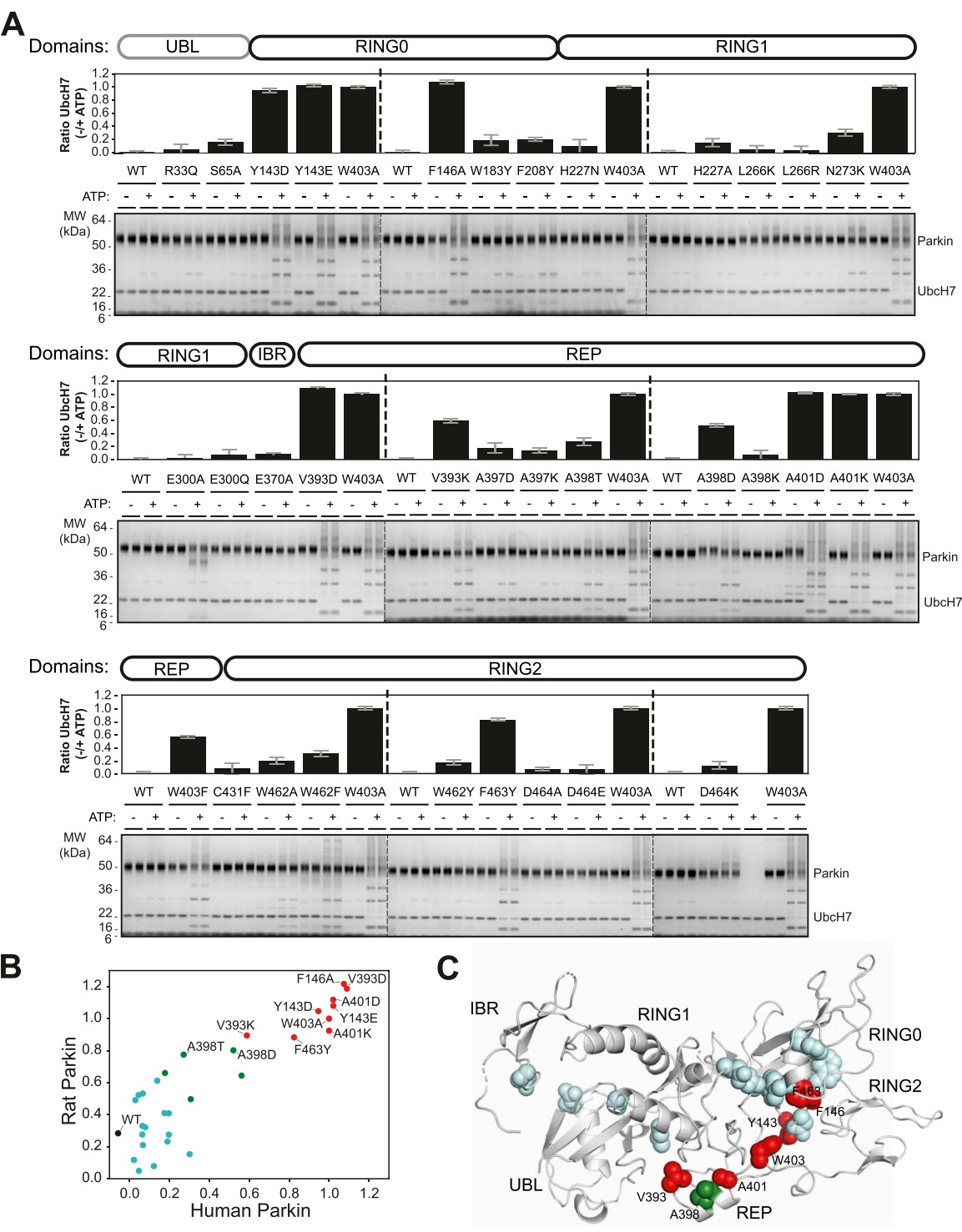

**Figure 2. Ubiquitination activity of Parkin mutants.**
**(A)** Human Parkin mutants were incubated with ATP, Ub, E1, and UbcH7 at 37°C for 1 h in the presence or absence of ATP. Reactions were resolved on 4–12% Bis-Tris gels followed by Coomassie blue staining (bottom). Activity was determined by measuring the proportion of UbcH7 band intensity lost upon addition of ATP. The bar chart (top) shows the mean depletion from all reactions on each Parkin mutant with error bars indicating the SD (n = 2). **(B)** Relative activities of human and rat Parkin mutants normalized to the W403A mutant. Dots are coloured by relative activity (red, high; green, medium; cyan, low). **(C)** The structure of auto-inhibited rat Parkin (PDB 4ZYN), with spheres indicating the position of mutated amino acids and the colours indicating the relative activity.

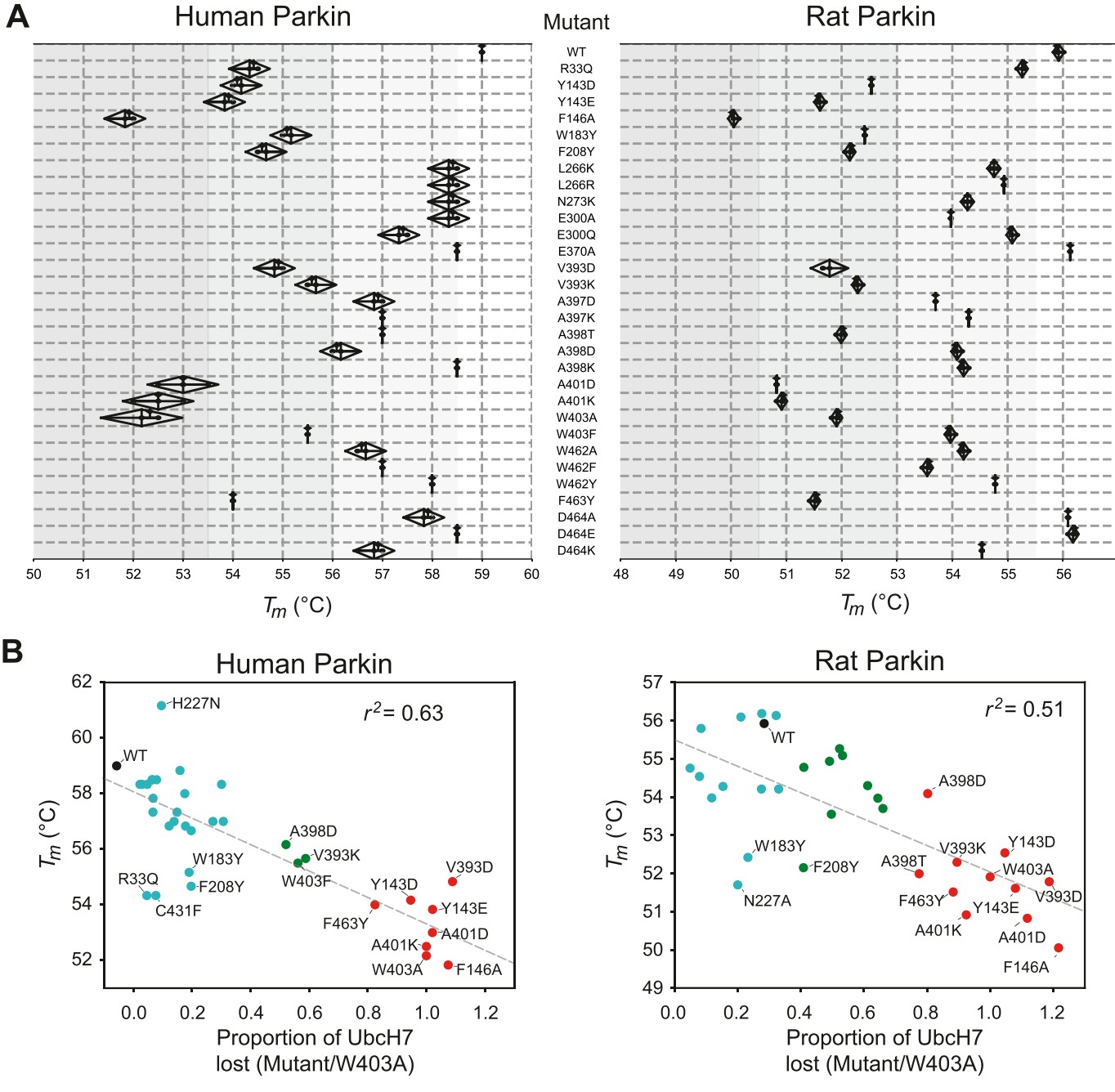

**Figure 3. Thermal stability of Parkin mutants and correlation with E3 ligase activity.**
**(A)** Thermal shift assays for human Parkin (left) and rat Parkin (right). For each mutant, the melting temperature ($T_m$) was determined by measuring the change in SYPRO-orange fluorescence intensity with temperature. The $T_m$ is the temperature which gives the maximum slope of the temperature against fluorescence. Measurements were made in triplicates, with the mean, maximum, and minimum $T_m$ from human and rat Parkin displayed in the above kite plots. **(B)** Scatter plots of the $T_m$ correlated with the E3 ligase activity, measured as loss of UbcH7 and normalized to W403A, for both human (left) and rat (right) Parkin mutants. The dashed line is a linear regression of all the data points, with the correlation coefficient $r^2$ indicated. Dots are coloured by relative activity (red, high; green, medium; cyan, low) as in Figs 2B and C.

we performed a correlation analysis between the two quantities (Fig 3B). The results for both rat and human Parkin showed an inverse correlation between the $T_m$ and the relative ubiquitination activity (Pearson correlation $r^2$: 0.63 in human, 0.51 in rat). Furthermore, if we restrict the analysis to mutations located at the RING0:RING2 or REP:RING1 interfaces, the correlation is even more

significant ($r^2$: 0.89 in human, 0.80 in rat), whereas there is no correlation for residues located outside this region (Fig S2). For example, mutations at Trp183, Asn227, and Glu300, which are all interacting with each other at the RING0:RING1 interface (Fig 1B), led to destabilization but no activation. Therefore, we conclude that activation is achieved specifically by reducing interdomain

interactions that lead to dissociation of the REP (necessary for E2 binding) and the RING0 domain (necessary for thioester transfer on the catalytic RING2 domain).

### Interplay between activating mutations and Parkin phosphorylation

To gain insights into the mechanism of activation of the Parkin mutants, we next compared their activity and stability to the phosphorylated form, which is activated via binding of the phospho$^{Ser65}$-Ubl and ACT to the RING0 domain. We thus performed in vitro phosphorylation of Parkin with recombinant insect PINK1 and purified the phosphorylated protein (p$^{S65}$-Parkin), which we used in ubiquitination assays to compare with activating mutations. For human Parkin, we added the substrate Miro to the ubiquitination reaction and found that p$^{S65}$-Parkin ubiquitinates Miro, and UbcH7, to a greater extent than the activating mutants W403A, A401K, and V393D (Fig 4A). For rat Parkin, we also find that p$^{S65}$-Parkin is more active than W403A, A401K, V393D, and A398T (Fig 4B). In thermal shift assays, we find that human p$^{S65}$-Parkin is mildly destabilized compared with apo Parkin, whereas rat p$^{S65}$-Parkin is destabilized to a greater extent (Fig 4C). Overall, these results suggest that although thermal stability predicts the relative activity of mutants, it does not correlate with the level of activity observed in the phosphorylated form.

Our ubiquitination assays also revealed that the Y143D and Y143E phospho-mimetic mutations activate Parkin in vitro (Figs 2 and S1). This result was surprising, given that two groups previously reported that phosphorylation of Tyr143 leads to the inactivation of the ligase activity (Ko et al, 2010; Imam et al, 2011). Given that p$^{S65}$-Parkin is the endogenous active form of Parkin in cells, we next asked whether phospho-mimetic mutations in Tyr143 could inhibit the activity of p$^{S65}$-Parkin. We thus pre-phosphorylated WT, Y143E, and W403A rat Parkin and then conducted autoubiquitination assays (Fig 4D). We found that both phosphorylated Y143E and W403A have enhanced activities compared with their non-phosphorylated forms, implying that the Y143E mutation does not inhibit pS65-mediated activation and that the activation mutants can be further activated by phosphorylation. Because glutamic acid lacks the aromatic group of a phospho-tyrosine, we next sought to test Tyr143 phosphorylation by c-Abl directly. However, c-Abl was unable to phosphorylate Parkin to a significant degree (Fig S3A). Tryptic digest and LC–MS/MS analysis revealed no significant phosphorylation of tyrosine residues in Parkin (Fig S3B), but extensive phosphorylation of tyrosine residues in c-Abl (Fig S3C), demonstrating that the enzyme is active. We then posited that ubiquitin or Parkin Ser65 phosphorylation could facilitate Tyr143 phosphorylation. We thus combined c-Abl with PINK1 and/or pUb and performed in vitro phosphorylation assays but found that c-Abl could not induce additional phosphorylation of Parkin (Fig S3A). We therefore incorporated the non-natural phospho-tyrosine mimetic p-carboxymethylphenylalanine at position Tyr143 by using the amber stop codon and modified tRNA and tRNA synthase (Xie et al, 2007). Strikingly, the Y143X mutation increased the autoubiquitination activity of Parkin, and similarly to the Y143E mutation (Fig 4E). Overall, these finding indicates that the addition of a negative

charge at Tyr143, at the intersection of RING0, RING1 and the C-terminus, results in the release of autoinhibition in Parkin.

### Mutation of Trp403 abrogates binding to the PRK8 monoclonal antibody

To aid in downstream studies, we next determined the best antibody for monitoring levels of Parkin mutants. We therefore performed SDS–PAGE analysis of all recombinant rat Parkin mutants and immunoblotted them against two antibodies. The first is the widely used monoclonal mouse antibody PRK8, directed towards the C-terminus of Parkin (Pawlyk et al, 2003). The second one is an in-house polyclonal sheep antibody that recognizes the N-terminus of Parkin (S229D; https://mrcppureagents.dundee.ac.uk/reagents-view-antibodies/589302). The PRK8 antibody robustly detected all mutants except W403A and W403F, implying that Trp403 is a key element of the antibody epitope (Fig S4). In contrast, the N-terminal Parkin antibody recognized all mutants equally well including W403A and W403F (Fig S4).

### Mitophagy assay identifies activating mutants that can rescue S65A Parkin

We next sought to determine whether mutations that increase Parkin ubiquitination activity in vitro also increase mitochondrial depolarization-dependent mitophagy in human cultured cells. Indeed, we had previously observed that the W403A and F146A Parkin mutants recruit faster to the mitochondria and increase mitophagy, as measured using the pH-sensitive fluorescent reporter mito-Keima (Katayama et al, 2011; Tang et al, 2017). We thus introduced 11 activating mutations in GFP-Parkin (human) for transient transfection in U2OS cells stably expressing ponasterone-inducible mito-Keima. Novel mutations that showed the greatest increase in activity in vitro were chosen (Y143E, Y143D, V393D, A401D, A401K, F463Y), and V393K, W403A, D464K, and E300A for comparison. All mutants expressed to similar levels and showed no accelerated degradation in the presence of the translation inhibitor cyclo-heximide, suggesting they do not undergo rapid autoubiquitination in cells (Fig S5). After GFP-Parkin transfection, mitophagy was induced with CCCP for 4 h and quantified using a dual excitation ratiometric pH measurement, with fluorescence-activated cell sorting to select GFP-Parkin positive cells. Cells that have a higher 561 nm (pH 4) to 407 nm (pH 7) ratio are considered to be positive for mitophagy (Figs 5 and S6). The results show that only the V393D and A401D mutants, in addition to W403A, show a significant increase in mitophagy.

We next determined which of those mutations would rescue the S65A mutant, which cannot be phosphorylated by PINK1. We had previously observed that the F146A and W403A mutations could rescue the S65A mutation (Tang et al, 2017). We therefore combined activating mutations with the S65A in the same protein (in *cis*) and measured normalized mitophagy. We included the mutants that showed a significant elevation of mitophagy (V393D, A401D, W403A), and Y143D/E, A398T, A401K, and F463Y mutants because they all increased Parkin's ubiquitination activity in vitro in rat and/or human Parkin (Fig 2A). The results show that there are only

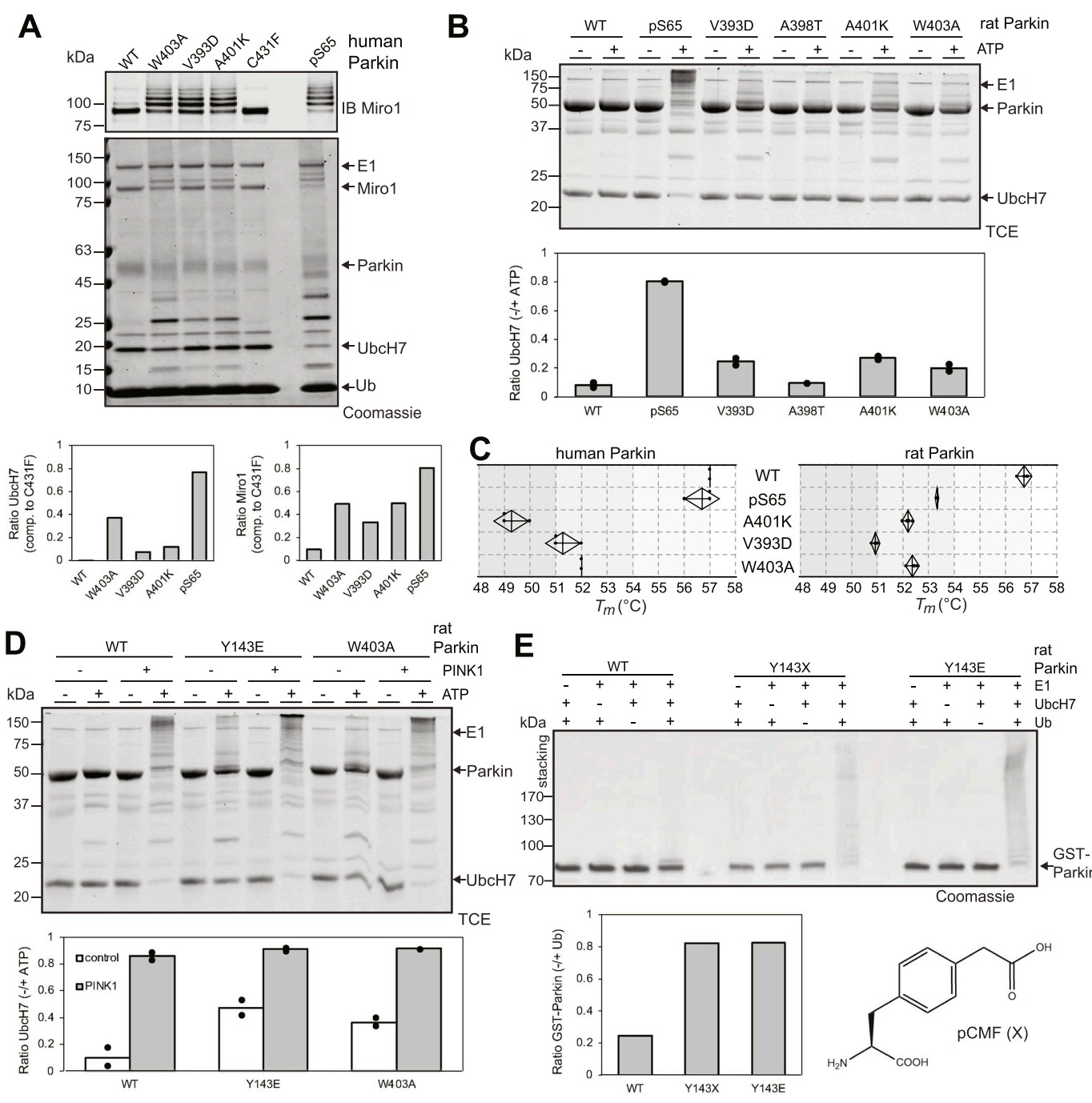

**Figure 4. Ubiquitination activity of Parkin Tyr143 phospho-mimetic mutants.**
**(A)** Ubiquitination assay of human Parkin mutants and p[S65]-Parkin incubated with ATP, Ub, E1, UbcH7, and the substrate Miro1 at 37°C for 30 min. Reactions were probed by SDS–PAGE and immunoblotting against Miro1 (top) or Coomassie blue staining (bottom). Activity was quantified by the loss of UbcH7 and Miro1 unmodified bands by densitometry and compared with the condition with the inactive mutant C431F (n = 1). **(B)** Ubiquitination assay of rat Parkin mutants and p[S65]-Parkin incubated with ATP, Ub, E1, and UbcH7 at 37°C for 2 h. Reactions were probed by SDS–PAGE and proteins visualized by fluorescence using trichloroethanol and UV light. Activity was quantified by the loss of unmodified UbcH7 by densitometry and compared with the condition with no ATP (n = 2). **(C)** Thermal shift assays for human Parkin (left) and rat Parkin (right), comparing pS65-Parkin with mutants, same as in Fig 3A. **(D)** Ubiquitination assay of rat Parkin mutants pre-incubated 30 min at 37°C with ATP and GST-TcPINK1, and then with Ub, E1, and UbcH7 at 37°C for 2 h. **(B)** Reactions were quantified as described in (B) (n = 2). **(E)** 30 min autoubiquitination assay with recombinant GST-fusion rat Parkin WT, Y143E, and Y143X, where X is pCMF. The chemical structure of the phospho-tyrosine mimetic amino acid p-carboxymethyl phenylalanine (pCMF) is shown below. Reactions were resolved by SDS–PAGE and proteins stained with Coomassie blue. Activity was quantified by the loss of unmodified GST-Parkin band by densitometry and compared with the condition with no ubiquitin (n = 2).

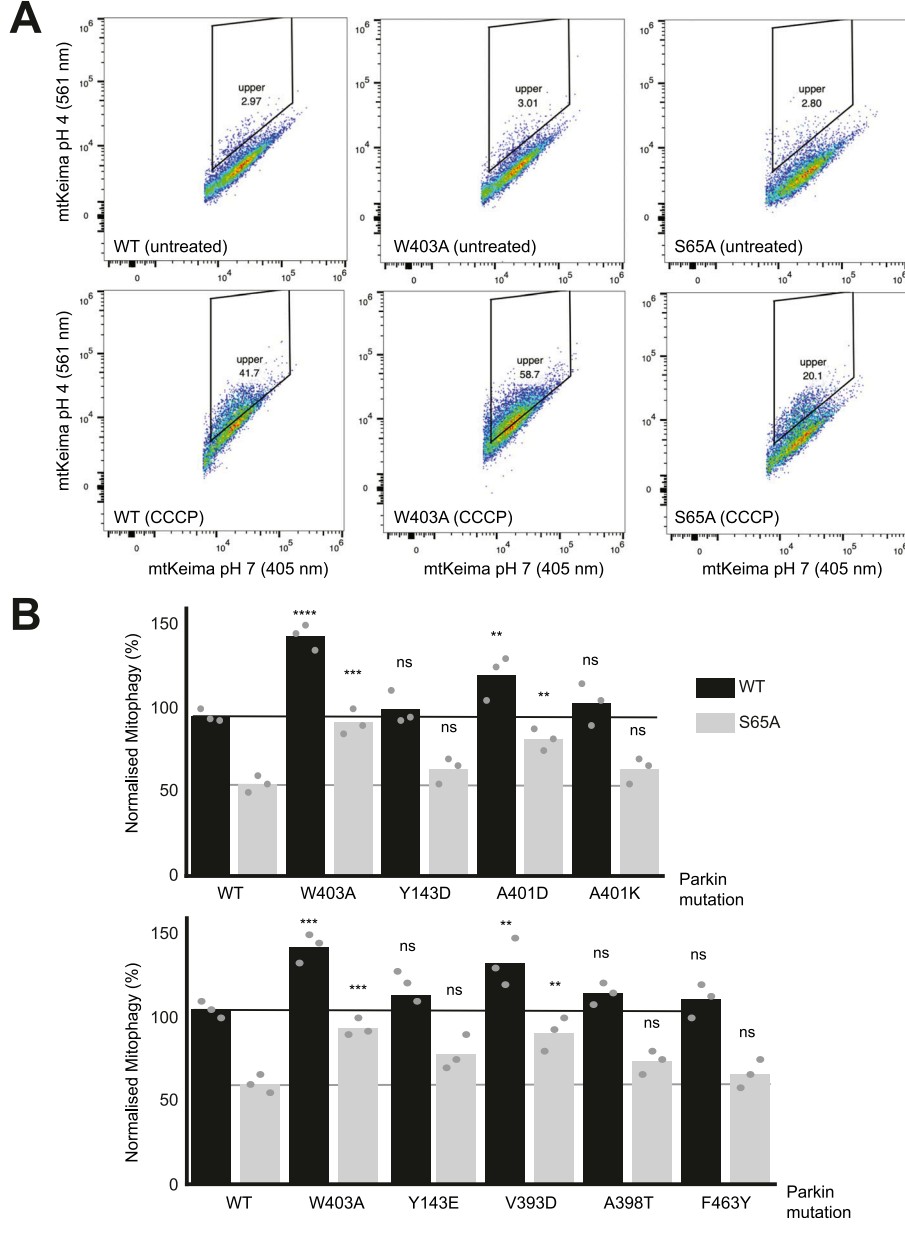

**Figure 5. The effect of Parkin mutations on mitophagy.**
**(A)** Representative FACS profiles to quantify mitophagy using mito-Keima. Assays were performed in U2OS cells stably expressing mito-Keima and transiently transfected with GFP-tagged human Parkin mutants. Mitophagy was detected by FACS of CCCP-treated (20 $\mu$M for 4 h) U2OS cells containing mitochondrially targeted mKeima (mt-Keima) and transiently expressing GFP-parkin. Cells in the upper quadrant (% indicated) are considered positive for mitophagy. **(B)** Bars indicate the percentage level of mitophagy, normalised to WT, for human Parkin mutants, introduced in *cis* in both WT (black) and S65A (grey) backgrounds. The horizontal lines indicate the WT level (dark grey) and S65A level (pale grey line). One-way ANOVA with Dunnett's post hoc tests (n = 3), *$P < 0.05$; **$P < 0.01$; ***$P < 0.001$; ****$P < 0.0001$; ns, not significant.

three mutants that can rescue S65A: V393D, A401D, and W403A (Fig 5B).

## Discussion

In this study, we engineered 31 mutations across all interdomain interfaces of Parkin including the Ubl:RING1, RING0:RING1, REP: RING1, and RING0:RING2 interactions. This led to the identification of 11 mutations that were capable of activating Parkin in both human and rat species in vitro (Figs 2 and S1). Notably, all mutations were localised near the RING0:RING2 or REP:RING1 interfaces and reduced the thermal stability of Parkin (Fig 3). On the other hand, mutations that destabilised the Ubl, IBR or RING0:RING1 interaction

had generally minor or no effect on activation, regardless of their impact on the thermal stability. For instance, mutations in Trp183, Phe208 or Glu300, at the RING0:RING1 interface, destabilise Parkin to various degrees, but do not activate Parkin E3 ligase activity. The mutation R33Q, which has been shown to reduce the thermal stability of the isolated Ubl domain (Safadi et al, 2011), also reduces the overall Parkin stability, but yet does not activate the ligase. In other words, reduced stability alone does not predict activation, but all activating mutations destabilise apo Parkin. This explains the stronger inverse correlation between $T_m$ and ligase activity observed for mutations at the REP:RING1 and RING0:RING2 interfaces alone (Figs 3 and S2). Furthermore, we find that p$^{S65}$-Parkin is more active than the most activating mutations such as V393D and W403A (Fig 4A and B), but yet, the thermal stability of p$^{S65}$-Parkin is not

further reduced (Fig 4C). This is not surprising, because the phospho-Ubl binds to the RING0 domain, thus replacing the RING2 domain; in other words, phosphorylation of Parkin leads to the loss of interactions (Ubl-RING1, REP-RING1, RING2-RING0), but also to the formation of new interactions (pUbl-RING0, ACT-RING0-pUbl), resulting in a change of thermal stability which is dependent on the net sum of these interactions.

Our data reinforce the current structural model of Parkin activation, which proposes that both the REP and RING2 must dissociate to allow E2~Ub binding and thioester transfer of Ub to Cys431 in RING2. Binding of phospho-Ubl to the RING0 domain stabilises this "open" configuration and thus displaces the chemical equilibrium towards the active form. Activating mutations also shift this equilibrium towards the "open" configuration and thus mimic the effect of Ubl phosphorylation. However, the mutations do not push the equilibrium as far as the Ubl phosphorylation; indeed, we find that the activating mutations Y143E and W403A can be further activated by PINK1-mediated phosphorylation, to a level comparable to phosphorylated WT form (Fig 4D). It should also be noted that activating mutations do not dispense Parkin from the necessary step of binding to pUb, which is essential for the recruitment of Parkin to mitochondria. Our activating mutations indeed do not induce mitophagy without addition of CCCP (Tang et al, 2017). Finally, it should be mentioned that we still lack a high-resolution structure of Parkin in the thioester transfer complex, which may reveal additional interactions mediated by the REP and/or RING2 that could be impacted by mutations aimed at releasing autoinhibitory interfaces.

Our observation that mutations Y143D, Y143E, and the phosphotyrosine mimetic Y143X increase Parkin's activity (Figs 1 and 4E and S2) was not surprising from a structural standpoint, because the side chain of Tyr143 in RING0 points towards the RING0:RING2 interface in the autoinhibited structure of rat Parkin (Trempe et al, 2013; Sauvé et al, 2015). The Y143E mutations can even be further activated by PINK1 phosphorylation (Fig 4D), implying that Tyr143 phosphorylation would not inhibit the active form. The interface notably includes Asp464 and the carboxy-terminal residue Val465 in RING2 (Fig 1). Thus, introduction of a negative charge at this position can lead to electrostatic repulsion and dissociation of the interface. It should be noted that the autoinhibited structure of human Parkin (Kumar et al, 2015) was determined from a construct with a deletion of the Ubl-RING0 linker that spans 84–143, and thus Tyr143 is not required to maintain the autoinhibited conformation. However, our results are in stark contrast to those obtained by two groups who reported that Tyr143 phosphorylation by c-Abl inhibits Parkin's activity in vitro and in vivo (Ko et al, 2010; Imam et al, 2011). The two groups notably showed that c-Abl can phosphorylate recombinant Parkin at Tyr143, and that the modification can be blocked by Gleevec (STI-571, imatinib), a selective Abl kinase inhibitor. c-Abl was proposed as a target for disease-modifying therapy for PD, with Abl inhibitors like nilotinib displaying neuroprotective activity against 1-methyl-4-phenyl-1,2,3,6-tetrahydropyridine (MPTP)-lesioned mice (Karuppagounder et al, 2014). However, we were not able to phosphorylate selectively Tyr143 in vitro using active c-Abl, under conditions where PINK1 phosphorylates Ser65 (Fig S3). Thus, our in vitro results suggest that c-Abl does not selectively phosphorylate Tyr143 in Parkin. However, we cannot exclude that

Parkin can be phosphorylated in cells by c-Abl with the assistance of some yet unknown factor, which may impact negatively Parkin's activity.

In previous work, we found that the F146A and W403A mutations accelerated Parkin recruitment to depolarized mitochondria and increased mitophagy flux (Tang et al, 2017). In contrast, the hyperactive mutant N273K, which disrupts the Ubl:RING1 interaction, exhibited slightly increased rates of mitochondrial recruitment and substrate ubiquitination but no alteration in mitophagy. These findings suggested that in cells, disruption of the REP:RING1 or RING0:RING2 interfaces but not the release of the Ubl domain were rate-limiting steps for Parkin activation on mitochondria and initiation of mitophagy. We therefore assessed whether any of the mutations that enhance Parkin activity in vitro could accelerate mitophagy in cells (Fig 5). We found that the V393D and A401D Parkin mutants were able to increase mitophagy flux similarly to the W403A mutation. On the other hand, the mutations Y143D, Y143E, A401K, and F463Y, which strongly activate Parkin in vitro (Fig 2), did not increase mitophagy (Fig 5). Although these mutants are not destabilised in cells (Fig S5), we cannot exclude that these mutations have additional effects on Parkin-mediated mitophagy that counteract any stimulating effect of its E3 ubiquitin ligase activity. We note that mutations of REP residues Val393 or Ala401 to aspartic acid increased mitophagy, whereas mutation of the same residues to lysine did not (Figs 5 and S6). Furthermore, we observed that the mutations A398T and A398D slightly increased Parkin's activity in vitro, whereas the A398K mutation did not (Fig 2). It therefore seems that introduction of a positively charged residue in the REP dampens activation, perhaps as a result of the REP making additional, yet unknown interactions in the thioester transfer complex with a charged E2~Ub conjugate.

Autosomal recessive mutations in Parkin represent the most frequent genetic cause of early-onset PD (Klein & Westenberger, 2012). The structure–function analysis of Parkin E3 ligase activity to date has been exemplar in understanding the mechanism of how PD mutations disrupt Parkin activation. We have notably reported the utility of the mitophagy assay in assessing 51 naturally occurring variants of Parkin and found that pathogenic variants exhibit severe mitophagy defects in cells, whereas clinically benign variants do not significantly impact mitophagy (Yi et al, 2019). Pathogenic variants could be attributed to the disruption of different features of Parkin, namely Ubl folding, PINK1 activation mechanism, catalytic activity, and zinc coordination. The two synthetic mutations F146A and W403A, and the naturally occurring variant V224A to a lesser extent, were able to rescue in *cis* defects in Ubl folding and most mutations in the PINK1 activation pathway, but not variants that disrupt catalytic activity or unfold the zinc-finger domains. For example, the Ubl folding mutant R42P, and the K161N and K211N mutations that prevent activatory pUbl binding to the RING0 domain, are rescued by the F146A and W403A mutations. This is consistent with the same activating mutations being able to rescue deletion of the Ubl or the S65A mutations (Tang et al, 2017). Here, we find that the V393D and A401D mutations, in addition to W403A, can also rescue the S65A mutation in *cis* (Fig 5B). On the other hand, the Y143D, Y143E, A398T, and F463Y mutations were not able to rescue the S65A mutation. The A398T mutation, which is a naturally occurring variant, did not strongly activate Parkin in vitro (Fig 2).

Likewise, the F463Y mutation had a more modest activating effect than the V393D and A401D mutations. Thus, it seems like the ability to rescue the S65A mutant in mitophagy correlates with in vitro E3 ligase activity, with the notable exception of the Tyr143 mutants. In future work, it will be interesting to test the ability of V393D and A401D to rescue pathogenic Parkin mutants in a manner similar to the W403A and F146A mutations.

Our findings add to accumulating data suggesting that destabilization of the REP:RING1 or RING0:RING2 interfaces can circumvent specific functional and pathogenic mutants of Parkin with therapeutic implications. The development of small molecule Parkin enzyme activators has remained extremely challenging in the field to date. Biogen has recently identified a number of compounds that act as positive allosteric modulators of Parkin (Shlevkov et al, 2022). These compounds sensitize Parkin to the activating effect of pUb or the W403A mutation but fail to enhance Parkin recruitment or mitophagy. The elaboration of further REP or RING0:RING2 domain-activating mutations and their ability to rescue Parkin function in cells adds further impetus to a review on how future screens of Parkin activators should be performed. Screening should be designed to specifically search for small molecules that mimic destabilisation of these intramolecular interfaces, while not affecting the interactions occurring in the active state. Finally, were a therapeutic to be developed that mimics activating mutations, it will then be important to test its ability to rescue different types of pathogenic Parkin mutations. Indeed, mutations that disrupt catalytic activity or zinc coordination are generally not rescuable, indicating the importance of stratifying Parkin mutant carriers that could benefit from this strategy.

## Materials and Methods

### Reagents

All mutagenesis was carried out by PCR site-directed mutagenesis using KOD polymerase (Merck). cDNA constructs for human Parkin mutant expression were amplified in *Escherichia coli* DH5$\alpha$ and purified using a NucleoBond Xtra Midi kit (#740420.50; Macherey-Nagel). All DNA constructs were verified by DNA sequencing, which was performed by The Sequencing Service, School of Life Sciences, University of Dundee, using DYEnamic ET terminator chemistry (Amersham Biosciences) on Applied Biosystems automated DNA sequencers. DNA for bacterial protein expression was transformed into *E. coli* BL21 DE3 RIL (codon plus) cells (Stratagene). cDNA plasmids (Table 1) and recombinant human Parkin proteins generated for this study are available to request through reagents website https://mrcppureagents.dundee.ac.uk/.

### Recombinant protein expression

#### Human Parkin

His$_6$-SUMO cleaved WT Parkin was expressed based upon the method (Kazlauskaite et al, 2014b). Briefly, plasmids were transformed in BL21 Codon Plus (DE3)-RIL *E. coli*, overnight cultures were prepared and used to inoculate 12 × 1 Liter of LB medium containing 50 $\mu$g/ml carbenicillin and 0.25 mM ZnCl$_2$. These were initially incubated at 37°C until the cultures reached an OD$_{600}$ of 0.4, the incubator temperature was lowered to 15°C, and once the cultures reached an OD$_{600}$ of 0.8–0.9, expression was induced by the addition of 25 $\mu$M IPTG. After overnight incubation (16 h), cells were pelleted by centrifugation (4,200$g$, 25 min), the media were removed, and the cell pellet was suspended in lysis buffer (50 mM Tris–HCl pH 7.5, 250 mM NaCl, 15 mM imidazole [pH 7.5], and 0.1 mM EDTA with 1 $\mu$M AEBSF and 10 $\mu$g/ml leupeptin added). Cells were burst by sonication and cell debris were pelleted by centrifugation (35,000$g$ for 30 min at 4°C) and the supernatant was incubated with Ni-NTA resin for 1 h at 5–7°C. Ni-NTA resin was washed five times in 7 × the resin volume of lysis buffer, and twice in 7 × the resin volume of cleavage buffer (50 mM Tris pH 8.3, 200 mM NaCl, 10% glycerol and 1 mM TCEP). Parkin was cleaved from the resin at 4°C overnight by the addition of a 1:5 mass ratio of His-SENP1 to total protein mass bound to the resin. After cleavage, Parkin was concentrated and further purified using size exclusion chromatography on a Superdex S200 column (16/600). Parkin was eluted after 80–90 ml, fractions were pooled, and concentrated purity was tested using SDS–PAGE. All the mutants could be concentrated to more than 1 mg/ml. The purity of each mutant was confirmed using SDS–PAGE. For phosphorylated human Parkin (pS65), His6-SUMO-fusion protein was captured on Ni-agarose, washed, and incubated with 5 mg of GST-TcPINK1 126–570 in the presence of 10 mM MgCl$_2$ and 2 mM ATP for 4 h at 27°C. The initial kinase and Mg-ATP were removed and replaced with fresh kinase and Mg-ATP for incubation over night at 27°C. The Ni-agarose was washed three times with wash buffer and Parkin was eluted and then dialysed in the presence of SENP1. The protein was purified further by chromatography on a Superdex 200 as previously described (Stevens & Muqit, 2021).

#### Rat Parkin

*Rattus norvegicus* full-length Parkin DNA was codon-optimized for *E. coli* expression and subcloned into pGEX6P-1 (DNAexpress). PCR mutagenesis was used to generate parkin single-point mutants. Parkin expression was done in BL21 (DE3) *E. coli* cells using conditions previously described (Hristova et al, 2009; Trempe et al, 2013). The cells were grown at 37°C to OD ~1.0, IPTG and ZnSO$_4$ were added to 25 $\mu$M, and the cells were left growing overnight at 16°C. The cells were pelleted by centrifugation (3,000$g$, 20 min), the media were removed, and the cell pellet was suspended in lysis buffer (50 mM Tris/HCl, 120 mM NaCl, 2.5 mM DTT, pH 8.0) with EDTA-free protease inhibitors cocktail (Roche). All proteins were purified using Glutathione-Sepharose 4B (Cytiva) and eluted with 20 mM reduced glutathione in 50 mM Tris/HCl, 120 mM NaCl, 2.5 mM DTT, pH 8.0. Eluted proteins were cleaved overnight with 3C protease followed by gel filtration on Superdex 200 16/60 (GE Healthcare) in 20 mM Tris–HCl, 120 mM NaCl, 2 mM DTT, pH 8.0. All the mutants could be concentrated to more than 1 mg/ml. The purity of each mutant was confirmed using SDS–PAGE. For phosphorylated rat Parkin (pS65), 49 $\mu$M purified WT cleaved Parkin was incubated with 10 $\mu$M GST-TcPINK1 (121–570), and 5 mM ATP for x min at 30°C. The reaction was monitored by intact mass spectrometry until completion of phosphorylation. The product was then injected on a Superdex 200 16/60 (GE Healthcare) in 20 mM Tris–HCl, 120 mM NaCl, 2 mM DTT, pH 8.0.

## Purification of Parkin Y143X mutant

The codon for Tyr143 was mutated to the TAG amber in the pGEX-6p1 rat Parkin plasmid and co-transformed with the pEvol-pCmF plasmid obtained from Peter G. Schultz, as described (Xie et al, 2007). Cells were grown overnight in LB medium (20 ml) with chloramphenicol (50 µg/ml)/ampicillin (50 µg/ml) from a single colony. This stock was used to inoculate 500 ml of M9 minimal media containing D-glucose (0.4% wt/vol), $NH_4Cl$ (0.1% wt/vol), $MgSO_4$ (4 mM), $CaCl_2$ (0.1 mM), thiamine/vitamin B1 (0.0002% wt/vol), $FeSO_4$ (4 µM), $ZnCl_2$ (4 µM) with chloramphenicol (50 µg/ml)/ampicillin (50 µg/ml), and grown at 37°C until OD600 ~0.5 is reached. Arabinose (0.2%) and pCmF (1 mM) were added and then the culture was grown for another hour. Temperature was reduced to 16°C and equilibrated for 30 min. IPTG (25 µM) and $ZnCl_2$ (25 µM) were added to induce an expression for 18 h (overnight) at 16°C. The GST-Parkin-Y143X protein was then purified as described above for rat Parkin.

## Purification of c-Abl and TcPINK1 kinases

Plasmids for human kinase c-Abl (pGEX-cAbl a.a. 83–531) and the YopH phosphatase and was obtained from the laboratory of John Kuriyan, as described (Seeliger et al, 2005). Both plasmids were transformed into BL21-DE3 cells in LB medium with streptomycin (50 µg/ml)/ampicillin (100 µg/ml). Cells were grown at 37°C until OD600 of 1.2 was reached and cooled for 1 h at 18°C with shaking. Expression was induced with 0.2 mM IPTG for 16 h (overnight) at 18°C. Cells were harvested by centrifugation at 7,000*g* at 4°C for 10 min and resuspended in 25 ml of cold TBS buffer (50 mM Tris–HCl, 500 mM NaCl, pH 8) with 5% glycerol and EDTA-free protease inhibitors cocktail (Roche). Cells were lysed by sonication and purified using Glutathione-Sepharose 4B (Cytiva) and eluted with 20 mM reduced glutathione in TBS. Eluted proteins were resolved by gel filtration on Superdex 200 16/60 (Cytiva) in TBS pH 7.5. GST-TcPINK1 (a.a. 121–570) was purified as described previously (Rasool et al, 2018), using a method identical to that described for rat Parkin above, with the exception that no $ZnCl_2$ was added in the medium.

## Parkin ubiquitylation assay

### Human Parkin

In vitro ubiquitylation assays were performed using recombinant proteins purified from *E. coli* unless stated otherwise. 1 µM of WT or mutant Parkin was incubated with the ubiquitin master mix (50 mM Tris–HCl pH 7.5, 10 mM $MgCl_2$, 2 mM ATP, 0.12 µM His-Ub E1 expressed in Sf21 insect cells, 1 µM human UbE2L3, 50 µM Flag-ubiquitin) to a final volume of 30 µl and the reaction was incubated at 37°C for 30 min in a thermo shaker at 1,000 rpm. Reactions were terminated by the addition of 4 × LDS loading buffer. 5 µl of the final reaction volumes were resolved using SDS–PAGE on a 4–12% Bis-tris gels in MOPS buffer and stained by incubating with a Coomassie stain (Instant Blue) overnight at room temperature. The stain was washed off using warm water until the gel background had cleared, approximately three washes. Gels were imaged using the Licor Odyssey Clx and band intensities were measured using Image Studio Lite. The proportion of UbcH7 depleted was measured by dividing the band intensity of the reaction in the presence of ATP by the band intensity in the absence of ATP and taking this value away from one. The assay was repeated twice, with the mean UbcH7 levels estimated by densitometry (Fig 2). For the assay showed in Fig 4A, the same reaction conditions were used, with the addition of 0.5 µM recombinant Miro1 purified as described (Kazlauskaite et al, 2014a). Samples (5 µl of the final volume) were resolved using SDS–PAGE on 4–12% Bis-Tris gels in MOPS buffer and transferred to nitrocellulose membranes. Membranes were blocked with 5% milk powder in TBS + 0.1% Tween 20 (TBS-T) for 1 h at room temperature then immunoblotted using an anti-Miro antibody (Abnova H00055288-M01) in 5% BSA/TBS-T at 4°C overnight. Protein bands were detected by blotting against secondary antibodies labelled with 800 or 680 nm fluorophores in TBS-T for 1 h at room temperature and imaged using LiCor.

### Rat Parkin

Ubiquitination assays were performed for 2 h at 37°C. Untagged Parkin (WT and mutant) at 4 µM was incubated with 200 nM E1, 2 µM UbcH7, 100 µM ubiquitin, 0.5 mM TCEP, and 10 mM ATP in the presence of 50 mM Tris–HCl pH 7.5 and 5 mM $MgCl_2$. Reactions were stopped with addition of SDS–PAGE sample buffer, resolved on a 12.5% acrylamide Tris/tricine gel containing 0.5% trichloroethanol, and imaged by fluorescence on a Gel Doc XR + Imaging System after UV exposure, (Bio-Rad), as described (Ladner-Keay et al, 2018). The assay was repeated twice, with the mean UbcH7 levels estimated by densitometry (Fig S1). For the assay showed in Fig 4B, the same conditions were used. For the assay showed in Fig 4D and 16 µM Parkin was incubated with 0.1 µM recombinant GST-TcPINK1 and 20 mM ATP for 30 min at 37°C, followed by addition of 200 nM E1, 2 µM UbcH7, 100 µM ubiquitin, 0.5 mM TCEP in the presence of 50 mM Tris–HCl pH 7.5 and 5 mM $MgCl_2$ (final Parkin concentration 4 µM) and incubation at 37°C for 2 h. For the assay with Parkin-Y143X and Y143E (Fig 4E), 1 µM GST-parkin was incubated at 37°C in 50 mM Tris/HCl, 50 mM NaCl, 0.5 mM DTT, pH 7.5, 2 mM ATP, 10 mM $MgCl_2$, 40 nM E1, and 2 µM UbcH7, for 60 min. Products were resolved by Tris-glycine SDS–PAGE and the gels stained with Coomassie blue.

## Thermal shift assays

### Human Parkin

Master mixes containing 1 × SYPRO Orange and 0.06 mg/ml Parkin in the final protein buffer (50 mM Tris pH 8.3, 200 mM NaCl, 10% glycerol, and 0.5 mM TCEP) were made up for each Parkin mutant. 50 µl of the master mix was aliquoted into three wells of a 96-well PCR plate. The assay was performed using a qPCR machine, for the assay a temperature gradient from 20–90°C was applied in 0.5°C steps with samples incubated at each step for 1 min. The intensity of SYPRO Orange emission (570 nm) was recorded at each temperature step. The Tm was determined by taking the first derivative of the SYPRO Orange emission curve and determining the temperature, this value reached its maximum.

### Rat Parkin

Stability of recombinant Parkin mutants was determined by fluorescence-based thermal shift assay. Parkin (wt and mutants) was mixed with SYPRO orange protein gel stain (Molecular

Probe–Life Technologies) to a final concentration of 0.5 mg/ml protein and 6× dye and heated from 10–70°C with fluorescence reading every 0.1°C (4–5 s) in a QuantStudio 7 Flex Real-Time PCR System (Life Technologies). Protein thermal melting curve were generated using the Protein Thermal Shift software. Four replicates were performed. The Tm was determined by taking the first derivative of the SYPRO Orange emission curve and determining the temperature, this value reached its maximum.

### Phosphorylation assay and mass spectrometry

Rat Parkin WT (5 $\mu$M) was incubated with 1 $\mu$M GST-cAbl and/or GST-TcPINK1, and/or 5 $\mu$M recombinant pUb, for 30 min at room temperature with 2 mM ATP in 50 mM Tris–HCl pH 7.5 and 5 mM MgCl$_2$. Reaction samples were diluted in denaturing buffer (3 M urea, 25 mM TEAB pH 8.5, 0.5 mM EDTA) and reduced using 2 mM TCEP for 10 min at 37°C, followed by alkylation with 50 mM chloroacetamide for 30 min at room temperature in the dark. Chloroacetamide was used to avoid iodoacetamide-induced modification of lysine side chains that mimic ubiquitination (Nielsen et al, 2008). Samples were diluted with 50 mM TEAB pH 8.5 to 1 M urea and digested with 0.5 $\mu$g trypsin (Sigma-Aldrich) for 3 h at 37°C. Digested peptides were purified using C18 Spin Columns (Thermo Fisher Scientific) and resuspended in 0.1% formic acid. Peptides (0.5 $\mu$g) were captured and eluted from an Acclaim PepMap100 C18 column with a 2 h gradient of acetonitrile in 0.1% formic acid at 200 nl/min. The eluted peptides were analysed with an Impact II Q-TOF spectrometer equipped with a Captive Spray nanoelectrospray source (Bruker). Data were acquired using data-dependent automatic tandem mass spectrometry (auto-MS/MS) and analysed with MaxQuant using a standard search procedure against a custom-made FASTA file including rat Parkin, GST-Abl, and GST-TcPINK1. Methionine oxidation and Ser/Thr/Tyr phosphorylation were included as variable modifications. Cysteine carbamylation was included as fixed modification. For intact mass spectrometry, reaction samples were diluted in 1% formic acid and 1 $\mu$g was injected on a Waters C4 BEH 1.0/10-mm column and washed 5 min with 4% acetonitrile, followed by a 10-min 4–90% gradient of acetonitrile in 0.1% formic acid, with a flow rate of 40 $\mu$l/min. The eluate was analyzed on a Bruker Impact II Q-TOF mass spectrometer equipped with an Apollo II ion funnel electrospray ionization source. Data were acquired in positive-ion profile mode, with a capillary voltage of 4,500 V and dry nitrogen heated at 200°C. Spectra were analyzed using the software DataAnalysis (Bruker). The multiply charged ion species were deconvoluted at 5,000 resolution using the maximum entropy method.

### Immunoblots

Parkin WT and mutants were immunoblotted with two Parkin antibodies. One ug protein was loaded on 10% acrylamide Tris-glycine gel followed by transfer to nitrocellulose membranes. The membranes were blocked followed by overnight incubation at 4°C with primary antibodies: C-terminal Parkin antibody (PRK8; Cell Signaling, 1:40,000 dilution) and N-terminal Parkin antibody (https://mrcppureagents.dundee.ac.uk/reagents-view-antibodies/589302; sheep poly-clonal antibody S229D, 1:240 dilution). HRP-conjugated secondary antibody (HRP-link anti-mouse IgG; Cell Signaling, 1:10,000 dilution for C-terminal Parkin Ab and HRP-link anti-sheep IgG; Sigma-Aldrich, 1:10,000 dilution for N-terminal Parkin Ab) was incubated 1 h at RT and protein visualized using Clarity Western ECL Substrate (Bio-Rad).

### Mitophagy assay

Mitophagy was examined using a fluorescence-activated cell sorting-based analysis of mitochondrially targeted mKeima as previously described (Tang et al, 2017; Yi et al, 2019). Briefly, U2OS cells stably expressing an ecdysone-inducible mt-Keima were induced with 10 $\mu$M ponasterone A, transfected with GFP-Parkin (WT, S65A and/or indicated mutants in *cis*) for 24 h and treated with 20 $\mu$M CCCP for 4 h. For flow cytometry analysis, cells were trypsinized, washed, and resuspended in PBS before analysis on a Thermo Attune NxT cytometer (Thermo Fisher Scientific) equipped with 405, 488, and 561 nm lasers and 610/20, 620/15, 530/30, and 525/50 filters (NeuroEDDU Flow Cytometry Facility, McGill University). Measurement of lysosomal mitochondrially targeted mKeima was made using a dual-excitation ratiometric pH measurement where pH 7 was detected using the 610/20 filter and excitation at 405 nm and pH 4 using the 620/15 filter and excitation at 561 nm. For each sample, 50,000 events were collected, and single, GFP-Parkin–positive cells were subsequently gated for mt-Keima (cells within the upper gate). Data were analysed using FlowJo v10.1 (Tree Star). For statistical analysis, the data represent the average percentage of mitophagy from three independent experiments, and *P*-values were determined by one-way ANOVA with Dunnett's post-hoc tests were performed. *$P < 0.05$, **$P < 0.01$, ***$P < 0.001$.

### Cycloheximide pulse-chase assay

***Cycloheximide pulse-chase assay and Western blotting analysis***
24 h after transfection, $5 \times 10^5$ cells were treated with 100 $\mu$g/ml CHX for the indicated amounts of time. Cells were harvested and then lysed in 150 $\mu$l of lysis buffer (50 mm Tris, pH 7, 8% glycerol [vol/vol], 0.016% SDS [wt/vol], 0.125% $\beta$-mercaptoethanol [vol/vol], 0.125% bromophenol blue [wt/vol], 1 mm PMSF, and 1 $\mu$g/ml leupeptin). The samples were sonicated and then resolved by SDS–PAGE on 10% gels along with Precision Plus All Blue protein prestained standards (Bio-Rad). After SDS–PAGE, proteins were transferred onto nitrocellulose membranes (LI-COR Biosciences). The membranes were blocked with 2.5% fish skin gelatin (Truin Science) in 1× PBS with 0.1% Triton X-100, probed with primary and secondary antibodies, and imaged with an Odyssey infrared imaging system using the manufacturer's recommended procedures (LI-COR).

# Supplementary Information

# Acknowledgements

This work was supported primarily by a research grant from the Michael J Fox Foundation (#14681 to J-F Trempe, MMK Muqit, EA Fon, TM Durcan,

W Springer), and a Wellcome Trust Senior Research Fellowship in Clinical Science (#210753/Z/18/Z to MMK Muqit), EMBO YIP Award (MMK Muqit), Canada Foundation for Innovation (#229792 to J-F Trempe), and Canada Research Chair grants (#950-229792 to J-F Trempe and #950-232176 to EA Fon). MA Eldeeb is a CIHR-Banting Fellow and supported by Parkinson Canada. W Springer is supported by NIH R01 NS085070. We are grateful to the sequencing service (School of Life Sciences, University of Dundee); Axel Knebel for expression and generation of ubiquitin reagents (MRC PPU); and MRC PPU Reagents and Services antibody teams (co-ordinated by James Hastie).

## Author Contributions

MU Stevens: formal analysis, investigation, visualization, methodology, and writing—review and editing.

N Croteau: formal analysis, investigation, methodology, and writing—review and editing.

MA Eldeeb: formal analysis, investigation, methodology, and writing—review and editing.

O Antico: data curation, formal analysis, and methodology.

ZW Zeng: investigation and methodology.

R Toth: resources and methodology.

TM Durcan: conceptualization and funding acquisition.

W Springer: conceptualization and funding acquisition.

EA Fon: conceptualization, supervision, funding acquisition, project administration, and writing—review and editing.

MMK Muqit: conceptualization, supervision, funding acquisition, project administration, and writing—review and editing.

J-F Trempe: conceptualization, data curation, formal analysis, supervision, funding acquisition, investigation, visualization, project administration, and writing—original draft, review, and editing.

## Conflict of Interest Statement

The authors declare that they have no conflict of interest.

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
