## [Reviewer comments · Life Science Alliance]

Life Science Alliance

Structure-based design and characterization of Parkin activating mutations

Michael Stevens, Nathalie Croteau, Mohamed Eldeeb, Odetta Antico, Zhi Zeng, Rachel Toth, Thomas Durcan, Wolfdieter Springer, Edward Fon, Miratul Muqit, and Jean-Francois Trempe

DOI: <https://doi.org/10.26508/lsa.202201419>

Corresponding author(s): Jean-Francois Trempe, McGill University

Review Timeline:

Submission Date:	2022-02-21
Editorial Decision:	2022-04-11
Revision Received:	2023-02-10
Editorial Decision:	2023-03-01
Revision Received:	2023-03-07
Accepted:	2023-03-08

Scientific Editor: Novella Guidi

Transaction Report:

April 11, 2022

Re: Life Science Alliance manuscript #LSA-2022-01419

Prof. Jean-Francois Trempe
McGill University
Pharmacology & Therapeutics
3655 Promenade Sir William Osler
McIntyre Building 1313
Montreal, Quebec H3G 1Y6
Canada

Dear Dr. Trempe,

Thank you for submitting your manuscript entitled "Structure-based design and characterization of Parkin activating mutations" to Life Science Alliance. The manuscript was assessed by expert reviewers, whose comments are appended to this letter. We invite you to submit a revised manuscript addressing the Reviewer comments.

Thank you for this interesting contribution to Life Science Alliance. We are looking forward to receiving your revised manuscript.

Sincerely,

B. MANUSCRIPT ORGANIZATION AND FORMATTING:

Reviewer #1 (Comments to the Authors (Required)):

In this paper, M. Stevens and N. Croteau et al., performed an unbiased screen for Parkin activating mutations, and out of 31 tested mutations, they report 11 mutations that increase Parkin activity in vitro, 3 of which are able to rescue mitophagy defects resulting from Parkin S65A mutant in a cell-based assay upon mitochondrial depolarisation with Parkin over-expression. Authors conclusion are well supported by the data showed in this work, which is of interest for the field, as understanding how Parkin is activated in the context of mitophagy might lead to generation of strategies to therapeutically intervene Parkinson's Disease patients with mutations in Parkin but also to further promote mitophagy in other pathological conditions. It would be of interest for the future to see how these mutations affect general animal physiology and mitophagy in more physiological conditions although this might be out of range for this paper.

Overall I think this is a nice paper within the scope of the journal, which adds valuable information to the field, and although I do not have any concerning comments there are several things that could be improved:

Major comments:

1. Authors show that mitophagy was increased upon mitochondrial depolarisation by CCCP in U2OS cell over-expressing GFP-tagged Parkin and the mitophagy reporter mt-Keima. It would have been better to use untagged Parkin but the use of GFP-tagged Parkin is justified as the strategy used to quantify mitophagy relies on FACS sorting of GFP positive cells to ensure Parkin expression in a given cell followed by gate sorting of mt-Keima signal 405/561.

However, there is not a single image of how the gating was adjusted or how the cells look like. It would be highly desirable to show a representative image of how mitophagy is increased (or not) with each Parkin mutant. This gives an easy and direct visual effect that is easier for the reader to understand straight away.

Minor comments:

1. In Figure 1A it is not clear where the REP and RING2 domains are in the activated diagram of Parkin.

2. In Figure 2A the meaning of the Y axis of the upper graphs is well explained in the figure legend, but it would be appropriate to include a label directly on the graph.

3. Although how cells were treated and how mitophagy was measured is well stated in the methods section it is not well explained in the results section nor in the figure legend of Figure 5. The conditions of how mitophagy was induced and quantified needs to be clearly stated.

4. It is not very clear whether in the rescue experiment of S65A, Parkin harbouring different mutations was co-expressed in addition to S65A-Parkin or mutations were combined.

5. It would be worthwhile to explain the revised version of the proposed/updated/actual Parkin activation mechanism in the results/discussion, i.e. is pUb binding necessary for parkin activating mutations?

6. There is a minor grammatical error in the first paragraph of the intro "an E3 ubiquitin (Ub) ligase that mediateS a mitochondrial quality control pathway "

Reviewer #2 (Comments to the Authors (Required)):

In their work Stevens, Croteau and Eldeeb et al. characterize the in vitro and in cellulo activity of an extensive collection of Parkin mutants designed to activate Parkin's E3 ubiquitin ligase activity. While in healthy individuals Parkin activation is achieved by PINK1-mediated phosphorylation of the Parkin Ser65 residue, neurodegenerative disease arises from deficiency in proper Parkin activity. Although several de-novo activating mutations have previously been reported to validate the Parkin activation mechanism proposed based on structural work, this study provides a more comprehensive dissection of each autoinhibitory interface using in vitro ubiquitination as a primary read-out. The authors report a strong positive correlation between the activity of equivalent mutants in human and rat Parkin as well as a strong inverse correlation between Parkin activity and protein stability, providing additional insights into the conservation and destabilization associated with Parkin activation. Lastly authors reveal that the requirements for the levels of Parkin activity in cellulo are more stringent than in vitro, showing that only the most highly active mutants were able to rescue the S65A mutation which disables PINK1-mediated Parkin activation. The key take-homes that Parkin activating mutations are destabilizing and cluster in the REP:RING1 and RING0:RING2 interfaces are strongly supported by the presented data. Of particular value to the community is that a number of groups came together not only to consistently and systematically analyze the Parkin activating mutations, but also to begin rigorously testing the many reported modes of Parkin activity modulation. A number of the conclusions drawn from this analysis would be strengthened by addressing the following points:

1) The authors conclude that activating mutants mimic the effect of Ubl phosphorylation. To support this claim, it is necessary to include a panel comparing the activity of purified Parkin phosphorylated on S65 by PINK1 with WT and the 'canonical' W403A activating mutant in Figures 2 and S1. Additionally, it may be informative to include a measurement of thermal stability of phospho-Parkin in Figure 3. Together these experiments should take no more than a few weeks. While destabilization of the activating mutants compared to unphosphorylated Parkin is expected, measuring activating mutant stability relative to phospho-Parkin would directly inform on the stabilizing benefits of formation of the RING0:phospho-Ubl:ACT interface in activated Parkin. This could inform on the viability of activating therapies targeting the REP:RING1 and RING2:RING2 interfaces.

2) The conclusion that Parkin phospho-mimetic mutants at position Y143 activate in vitro Parkin activity is based on comparing Y143X parkin activity to WT, unphosphorylated Parkin activity. Since the physiologically active form of Parkin is phospho-S65 Parkin, it is necessary to test the effect of the phosphomimetic substitutions at position 143 in the context phospho-S65 Parkin before making a direct comparison with previous in vivo studies. Similarly, these experiments should take no more than a few weeks. Y143 is positioned in close proximity of the RING0:phospho-Ubl:ACT activating interface, and may reasonably result in decreased phospho-S65 Parkin activity. The phospho-S65 Parkin active confirmation may also be a better substrate for c-Abl mediated phosphorylation.

3) The control top blot from Figure S5 appears to be missing, and is crucial to show that the heavily in vitro destabilized Parkin activating mutants are indeed as stable as WT Parkin in cells.

4) The authors find that while a number of Parkin mutants seem to show maximum reactivity at the one hour in vitro assay endpoint only the REP mutants W403A, A401D and V393D rescue mitophagy in cells, therefore the authors argue for the therapeutic benefits of targeting the REP:RING0 interface. However, the previously reported to rescue mitophagy F146A mutation is not included in Figure 5, and the only additional RING0:RING2 interface mutations tested are the conservative F463Y and less well buried D464K, both of which are less activating than F146A in vitro. To substantiate their strong claims about the benefits of therapeutically targeting the REP:RING0 interface specifically, the authors should include a more comprehensive analysis of the RING0:RING2 interface. A more stringent, shorter in vitro time course would help tease apart the most activating mutations and explain differences seen in Parkin mutant behaviour. Since these experiments may take several months, the authors may consider softening the statements made about the benefits of targeting the REP:RING0 interface in their concluding paragraph instead.

5) Although in the abstract the authors describe this study as a 'comprehensive mutational analysis to unbiasedly determine Parkin activating mutations', all mutations were rationally chosen. I believe there is interest in the field to develop an unbiased way of identifying and testing potential activating mutations in the wake of our understanding of the Parkin active form (e.g. by MD simulation), and would therefore ask the authors to modulate their opening statement about the comprehensive and unbiased scope of their study to encourage further work on the subject.

Thanks to the reviewers for their supportive comments and constructive suggestions. As a sidenote, we do intend to pursue these mutations in a more physiological context, but this will be published in a subsequent paper. Below, you will find our response to every point in red.

Reviewer #1 (Comments to the Authors (Required)):

In this paper, M. Stevens and N. Croteau et al., performed an unbiased screen for Parkin activating mutations, and out of 31 tested mutations, they report 11 mutations that increase Parkin activity in vitro, 3 of which are able to rescue mitophagy defects resulting from Parkin S65A mutant in a cell-based assay upon mitochondrial depolarisation with Parkin over-expression. Authors conclusion are well supported by the data showed in this work, which is of interest for the field, as understanding how Parkin is activated in the context of mitophagy might lead to generation of strategies to therapeutically intervene Parkinson's Disease patients with mutations in Parkin but also to further promote mitophagy in other pathological conditions. It would be of interest for the future to see how these mutations affect general animal physiology and mitophagy in more physiological conditions although this might be out of range for this paper.

Overall I think this is a nice paper within the scope of the journal, which adds valuable information to the field, and although I do not have any concerning comments there are several things that could be improved:

Major comments:

1. Authors show that mitophagy was increased upon mitochondrial depolarisation by CCCP in U2OS cell over-expressing GFP-tagged Parkin and the mitophagy reporter mt-Keima. It would have been better to use untagged Parkin but the use of GFP-tagged Parkin is justified as the strategy used to quantify mitophagy relies on FACS sorting of GFP positive cells to ensure Parkin expression in a given cell followed by gate sorting of mt-Keima signal 405/561.

However, there is not a single image of how the gating was adjusted or how the cells look like. It would be highly desirable to show a representative image of how mitophagy is increased (or not) with each Parkin mutant. This gives an easy and direct visual effect that is easier for the reader to understand straight away.

Indeed, untagged Parkin would be ideal, but we do rely on the GFP signal for fluorescence-cell sorting. We have added representative FACS profiles in the main figure 5A for WT, W403A, and S65A, and then a panel for all mutants in supplemental Figure S6. The gating was adjusted so that in untreated cells, the percentage of cells in the upper quadrant is close to zero and increases to >40% with WT Parkin treated with CCCP for 4 h. The same system was used and described in a previous publication

(Tang et al 2017) and the level of activity observed mirrors the pathogenicity of the PD mutation (Yi et al, Hum Mol Genet 2019).

Minor comments:

1. *In Figure 1A it is not clear where the REP and RING2 domains are in the activated diagram of Parkin.*

These domains were not observed in the crystal structures of activated Parkin, either because they were disordered or because they were part of the recombinant protein. We have added a note in the legend of Figure 1A to clarify this point.

2. *In Figure 2A the meaning of the Y axis of the upper graphs is well explained in the figure legend, but it would be appropriate to include a label directly on the graph.*

Agreed. This has been corrected in Figure 2, as well as Suppl. Figure S1

3. *Although how cells were treated and how mitophagy was measured is well stated in the methods section it is not well explained in the results section nor in the figure legend of Figure 5. The conditions of how mitophagy was induced and quantified needs to be clearly stated.*

Sentences were added to the results (p. 11) and the legend of Figure 5 to describe how mitophagy was induced and quantified.

4. *It is not very clear whether in the rescue experiment of S65A, Parkin harbouring different mutations was co-expressed in addition to S65A-Parkin or mutations were combined.*

They were combined. We have clarified the main text, figure legends and method sections to make it clear that the mutations are in "cis".

5. *It would be worthwhile to explain the revised version of the proposed/updated/actual Parkin activation mechanism in the results/discussion, i.e. is pUb binding necessary for parkin activating mutations?*

We have expanded a paragraph in the discussion (pp. 13-14) about the implication of our findings for the activation mechanism.

6. *There is a minor grammatical error in the first paragraph of the intro "an E3 ubiquitin (Ub) ligase that mediateS a mitochondrial quality control pathway "*

Corrected.

Reviewer #2 (Comments to the Authors (Required)):

In their work Stevens, Croteau and Eldeeb et al. characterize the in vitro and in cellulo activity of an extensive collection of Parkin mutants designed to activate Parkin's E3 ubiquitin ligase activity. While in healthy individuals Parkin activation is achieved by PINK1-mediated phosphorylation of the Parkin Ser65 residue, neurodegenerative disease arises from deficiency in proper Parkin activity. Although several de-novo activating mutations have previously been reported to validate the Parkin activation mechanism proposed based on structural work, this study provides a more comprehensive dissection of each autoinhibitory interface using in vitro ubiquitination as a primary read-out. The authors report a strong positive correlation between the activity of equivalent mutants in human and rat Parkin as well as a strong inverse correlation between Parkin activity and protein stability, providing additional insights into the conservation and destabilization associated with Parkin activation. Lastly authors reveal that the requirements for the levels of Parkin activity in cellulo are more stringent than in vitro, showing that only the most highly active mutants were able to rescue the S65A mutation which disables PINK1-mediated Parkin activation. The key take-homes that Parkin activating mutations are destabilizing and cluster in the REP:RING1 and RING0:RING2 interfaces are strongly supported by the presented data. Of particular value to the community is that a number of groups came together not only to consistently and systematically analyze the Parkin activating mutations, but also to begin rigorously testing the many reported modes of Parkin activity modulation. A number of the conclusions drawn from this analysis would be strengthened by addressing the following points:

1) The authors conclude that activating mutants mimic the effect of Ubl phosphorylation. To support this claim, it is necessary to include a panel comparing the activity of purified Parkin phosphorylated on S65 by PINK1 with WT and the 'canonical' W403A activating mutant in Figures 2 and S1. Additionally, it may be informative to include a measurement of thermal stability of phospho-Parkin in Figure 3. Together these experiments should take no more than a few weeks. While destabilization of the activating mutants compared to unphosphorylated Parkin is expected, measuring activating mutant stability relative to phospho-Parkin would directly inform on the stabilizing benefits of formation of the RING0:phospho-Ubl:ACT interface in activated Parkin. This could inform on the viability of activating therapies targeting the REP:RING1 and RING2:RING2 interfaces.

We have tested phospho-Parkin in both autoubiquitination and thermal shift assays, for both human and rat Parkin. For both species, we find that phospho-Parkin increases activity to a greater extent than all mutants, including W403A (new Figure 4A/B). This is consistent with previous observations from our group, which showed that phosphoSer65-Parkin has a higher affinity for Ubch7 than W403A (Sauvé et al 2015). In thermal shift assays, phosphorylation at Ser65 mildly destabilizes human Parkin, but had an effect comparable to W403A in rat Parkin (new Figure 4C). Thus, it appears that the correlation between thermal stability and activity only holds for mutants at the RING0-RING2 or REP:RING1 interface. This is not surprising, because the phospho-

Ubl binds to the RING0 domain, thus replacing the RING2 domain; in other words, phosphorylation of Parkin leads to the loss of interactions (Ubl-RING1, REP-RING1, RING2-RING0), but also to the gain of interactions (pUbl-RING0, ACT-RING0-pUbl), the net sum of which is slightly negative and depends on the species. In the case of the mutants, there is only a loss of interactions, which leads to a larger magnitude drop in stability. Finally, we also find that the activating mutants Y143E and W403A can be further activated by phosphorylation, which again emphasizes that the mutations do not fully activate Parkin. We expand on the implications of these findings in the discussion.

2) The conclusion that Parkin phospho-mimetic mutants at position Y143 activate in vitro Parkin activity is based on comparing Y143X parkin activity to WT, unphosphorylated Parkin activity. Since the physiologically active form of Parkin is phospho-S65 Parkin, it is necessary to test the effect of the phosphomimetic substitutions at position 143 in the context phospho-S65 Parkin before making a direct comparison with previous in vivo studies. Similarly, these experiments should take no more than a few weeks. Y143 is positioned in close proximity of the RING0:phospho-Ubl:ACT activating interface, and may reasonably result in decreased phospho-S65 Parkin activity. The phospho-S65 Parkin active confirmation may also be a better substrate for c-Abl mediated phosphorylation.

Unfortunately, we no longer have the reagents necessary to make the Y143X mutant again. But we were able to conduct autoubiquitination assay with the Y143E mutant, which was pre-phosphorylated with PINK1 (new Figure 4D). We also included WT and W403A for comparison. The results show unambiguously that phosphorylation by PINK1 further increases the activity of both Y143E and W403A, to a level comparable to phospho-WT. Thus, we can conclude that the phosphomimetic mutation Y143E does not inhibit pS65-Parkin.

To determine whether phospho-Parkin (or even pUb) could make Parkin a better substrate for c-Abl, we perform in vitro phosphorylation assays assessed by intact mass spectrometry (new Figure S3A). We could not detect a single additional phosphorylation site induced by c-Abl, whereas PINK1 readily phosphorylated both Parkin and Parkin:pUb. Under these conditions, c-Abl massively autophosphorylated, confirming that the enzyme was active (Figure S3B). We can therefore conclude that c-Abl does not phosphorylate Parkin, at least under these conditions.

Still, we cannot exclude that in cells or in vivo, Parkin might get phosphorylated by c-Abl at Y143. Thus, all results related to phosphorylation have been merged in the new Figure 4, and we have renamed this results section title to put more emphasis on the interplay between activating mutations and PINK1-mediated phosphorylation.

3) The control top blot from Figure S5 appears to be missing, and is crucial to show that the heavily in vitro destabilized Parkin activating mutants are indeed as stable as WT Parkin in cells.

We have added WT Parkin to the panel (new Figure S5)

4) *The authors find that while a number of Parkin mutants seem to show maximum reactivity at the one hour in vitro assay endpoint only the REP mutants W403A, A401D and V393D rescue mitophagy in cells, therefore the authors argue for the therapeutic benefits of targeting the REP:RING0 interface. However, the previously reported to rescue mitophagy F146A mutation is not included in Figure 5, and the only additional RING0:RING2 interface mutations tested are the conservative F463Y and less well buried D464K, both of which are less activating than F146A in vitro. To substantiate their strong claims about the benefits of therapeutically targeting the REP:RING0 interface specifically, the authors should include a more comprehensive analysis of the RING0:RING2 interface. A more stringent, shorter in vitro time course would help tease apart the most activating mutations and explain differences seen in Parkin mutant behaviour. Since these experiments may take several months, the authors may consider softening the statements made about the benefits of targeting the REP:RING0 interface in their concluding paragraph instead.*

We agree with the reviewer here. Our intent was not to suggest that the REP:RING1 interface was the only one that could be targeted, but rather to highlight that REP mutations can rescue defective mutant S65A in mitophagy. We have modified the concluding paragraph and abstract to make it clear that both interfaces could be targeted.

5) *Although in the abstract the authors describe this study as a 'comprehensive mutational analysis to unbiasedly determine Parkin activating mutations', all mutations were rationally chosen. I believe there is interest in the field to develop an unbiased way of identifying and testing potential activating mutations in the wake of our understanding of the Parkin active form (e.g. by MD simulation), and would therefore ask the authors to modulate their opening statement about the comprehensive and unbiased scope of their study to encourage further work on the subject.*

We agree again with the reviewer and have modified our abstract to better reflect the rationale behind our approach, and what should be done in the future.

March 1, 2023

RE: Life Science Alliance Manuscript #LSA-2022-01419R

Prof. Jean-Francois Trempe
McGill University
Pharmacology & Therapeutics
3655 Promenade Sir William Osler
McIntyre Building 1313
Montreal, Quebec H3G 1Y6
Canada

Dear Dr. Trempe,

Thank you for submitting your revised manuscript entitled "Structure-based design and characterization of Parkin activating mutations". We would be happy to publish your paper in Life Science Alliance pending final revisions necessary to meet our formatting guidelines.

- please upload your supplementary figures as single files and add a separate figure legend section to your main manuscript text
- please make sure that the author order in our system matches the author order in your manuscript

A. FINAL FILES:

B. MANUSCRIPT ORGANIZATION AND FORMATTING:

**Submission of a paper that does not conform to Life Science Alliance guidelines will delay the acceptance of your

manuscript.**

The license to publish form must be signed before your manuscript can be sent to production. A link to the electronic license to publish form will be sent to the corresponding author only. Please take a moment to check your funder requirements.

Sincerely,

Reviewer #1 (Comments to the Authors (Required)):

The authors satisfactorily addressed the points raised during the review process, therefore I find the paper ready to be published as it.

Reviewer #2 (Comments to the Authors (Required)):

The authors have addressed all my concerns and suggestions. I congratulate them on this piece of work, and am happy to happy to recommend the manuscript for publication.

March 8, 2023

RE: Life Science Alliance Manuscript #LSA-2022-01419RR

Prof. Jean-Francois Trempe
McGill University
Pharmacology & Therapeutics
3655 Promenade Sir William Osler
McIntyre Building 1313
Montreal, Quebec H3G 1Y6
Canada

Dear Dr. Trempe,

Thank you for submitting your Research Article entitled "Structure-based design and characterization of Parkin activating mutations". It is a pleasure to let you know that your manuscript is now accepted for publication in Life Science Alliance. Congratulations on this interesting work.

DISTRIBUTION OF MATERIALS:

Again, congratulations on a very nice paper. I hope you found the review process to be constructive and are pleased with how the manuscript was handled editorially. We look forward to future exciting submissions from your lab.

Sincerely,
